# Recent Research Progress in Indophenine-Based-Functional Materials: Design, Synthesis, and Optoelectronic Applications

**DOI:** 10.3390/ma16062474

**Published:** 2023-03-20

**Authors:** Shiwei Ren, Abderrahim Yassar

**Affiliations:** 1Zhuhai Fudan Innovation Institution, Guangdong-Macao In-Depth Cooperation Zone in Hengqin, Guangdong 519000, China; shiwei_ren@fudan.edu.cn; 2LPICM, Ecole Polytechnique, CNRS, Institut Polytechnique de Paris, 91128 Palaiseau, France

**Keywords:** quinoidal π-conjugated materials, OFETs, indophenine dyes

## Abstract

This review highlights selected examples, published in the last three to four years, of recent advance in the design, synthesis, properties, and device performance of quinoidal π-conjugated materials. A particular emphasis is placed on emerging materials, such as indophenine dyes that have the potential to enable high-performance devices. We specifically discuss the recent advances and design guidelines of π-conjugated quinoidal molecules from a chemical standpoint. To the best of the authors’ knowledge, this review is the first compilation of literature on indophenine-based semiconducting materials covering their scope, limitations, and applications. In the first section, we briefly introduce some of the organic electronic devices that are the basic building blocks for certain applications involving organic semiconductors (OSCs). We introduce the definition of key performance parameters of three organic devices: organic field effect transistors (OFET), organic photovoltaics (OPV), and organic thermoelectric generators (TE). In section two, we review recent progress towards the synthesis of quinoidal semiconducting materials. Our focus will be on indophenine family that has never been reviewed. We discuss the relationship between structural properties and energy levels in this family of molecules. The last section reports the effect of structural modifications on the performance of devices: OFET, OPV and TE. In this review, we provide a general insight into the association between the molecular structure and electronic properties in quinoidal materials, encompassing both small molecules and polymers. We also believe that this review offers benefits to the organic electronics and photovoltaic communities, by shedding light on current trends in the synthesis and progression of promising novel building blocks. This can provide guidance for synthesizing new generations of quinoidal or diradical materials with tunable optoelectronic properties and more outstanding charge carrier mobility.

## 1. Introduction

Organic electronic devices have gained enormous popularity over the past 30 years because of their promising tunable electronic properties, flexibility, low-cost, versatile functionalization, and processability. The critical component of these devices is the OSC material, which serves as the active layer and determines the performance of the device. Over the last decade, one of the driving forces of development within this field is synthesizing novel, high-performance building blocks and extending the library of organic materials for various optoelectronic and energy applications. However, most the reported works focus on development of hole-transport (p-type) OSCs, while less research has been conducted on electron-transport (n-type) OSCs materials. One main reason for the lack of high-performance n-type OSCs is the availability of π-conjugated building units with strong electron-deficiency necessary to ensure sufficiently deep-lying LUMO energy and to enable the fabrication of n-type OSCs with stable electron transport features. For electron transport materials, the majority can only undergo stable n-type transport under the conditions of nitrogen or vacuum, as electron carriers can be easily trapped by H_2_O and O_2_ in the environment during the manufacturing of the device. It is generally assumed that a LUMO energy level at −4.0 eV is a prerequisite to developing a stable electron transport OSC, as the presence of a high electron affinity to facilitate electron injection and environmental stability enables the acquisition of high-performance electron transport OFETs. Furthermore, the specific building blocks should have selective reaction sites that provide handles for insertion into the π-conjugated system and should be readily compatible with a broad range of chemical reactions [1]. Frontier molecular orbital (FMO) energy levels (both HOMO and LUMO energy levels) can be accurately can be accurately regulated by modifying them to match the work function of the electrode. Despite the impressive work on the synthesis of n-type OSCs, the further development of novel building blocks enabling the production of high-performance materials remains a serious challenge. The exploration in this area has been driven by the development of new applications that require specific molecular design, namely non-fullerene organic solar cells, organic thin film transistors (OTFT), organic electrochemical transistors (OECT), organic thermoelectric (TE) generators, etc.

According to all databases from Web of Science, accessed in the winter of 2022, the number of published works in the domain of n-type and OSCs from 2004 to 2022 has exponentially increased, indicating the importance and interest of such emerging fields. (Figure 1) As shown in Figure 1, there are approximately one hundred review articles published related to n-type OSCs, covering synthesis design, characterization, and device function. Indophenine dyes are a sub-family of quinoidal small molecules having an oxindole moiety as a terminal group. According to SciFinder data there are only 106 scientific publications related to indophenine molecules, and no review articles on this topic have been published in open literature. 

This review particularly highlights the recent advances in the emerging field of OSCs based on indophenine π-conjugated molecules, with particular emphasis on synthesis, characterization, device fabrication, and function. To this end, we analyse the relationship between the molecular structure (nature of building blocks, crystallinity, morphology, etc.) and the energy levels, air stability, as well as the charge carrier mobilities. Although there are some reviews on n-type OSCs and quinoidal π-conjugated systems for optoelectronic and energy applications, most of them cover only specific applications, and none of them cover the entire spectrum ranging from the molecular design, synthetic tactics to device performance.

## 2. Basic Background of Organic Electronic Devices and Assessment Parameters

In this section, we briefly outline certain organic electronic devices, which are the primary building blocks for various OSC-based applications, Figure 2. For a more detailed discussion, some excellent reviews focusing on this topic are recommended [2].

OFETs are not only the fundamental building blocks of flexible and large area electronic devices but are also a useful tool for measuring charge-carrier mobilities of newly OSCs, and for understanding, assessing charge transport behavior of OSCs. OFETs can be fabricated in different ways, with the most common constructions being the bottom-gate bottom-contact (BGBC) and bottom-gate top-contact (BGTC) structures [3]. For the first case, the source and drain electrodes are placed directly on the dielectric film, where most of the charge carriers are expected to be transported; whereas for the second case, the source and drain electrodes are positioned on top of the semiconducting layer, and the charge carriers must cross the semiconducting layer to arrive at the channel. Therefore, with those straightforward differences, it means that the device configuration is able to affect the extracted charge mobility [4]. OFETs can operate in two regimes: linear and saturation regimes. One figure of merit for OFETs is the charge mobility, which could be extracted by using OFET-equation in linear and/or saturation regimes. Although OFETs were already widely used to evaluate the charge transport characteristics in newly synthetized OSCs; nonetheless, it is challenging to compare charge transport parameters of different materials as the OFET-mobility is governed by several factors such as the OFET-configuration used, the contact resistances, the choice of dielectric through its surface morphology, and the morphology of active layer.

The TE effect is a straight conversion of temperature differences into electric voltages and vice versa. Research is focused on novel TE-materials due to increasing energy demand and pollution linked to human activities. It should be remarked that approximately two-thirds of all industrial energy consumption is lost in the form of waste heat [5]. Consequently, it becomes urgent to promote the recovery of this huge waste heat into electrical energy. During the previous ten years, the number of teams active in the research of organic-TE materials has increased significantly. Since they are lightweight, printable, and suitable for large area applications, both p- and n-type TE materials become necessary in practical applications. However, the efficiency of the TE is quite low. The performance of TE is commonly expressed in terms of TE figure of merit, ZT = σS ^2^T/k, whereby σ, S, T, and k, respectively, represents electrical conductivity, Seebeck coefficient, absolute temperature, and thermal conductivity [6]. The optimal TE-material should have either higher Seebeck coefficient for improved energy conversion, higher electrical conductivity for decreased joule heating, or lower thermal conductivity to conserve the temperature gradient. One of the challenges in the field of TE is the strong interdependence between the σ, S and κ, with optimization of any one of them causing a negative impact on at least one of the others.

In addition to OFET and TE applications, n-type OSCs have also attracted considerable interest as alternative acceptors for non-fullerene solar cells [7] and electron-transporting materials for p-i-n perovskite. Organic and perovskite solar cells offer many benefits: flexibility, printability, low weight, low cost, fashionable designs, the ability to be manufactured by roll-to-roll production, etc. For both types of solar cells, fullerenes and their derivatives have been broadly applied as an acceptor for OPV. However, they suffer from a number of drawbacks including weak light absorption in the visible-near infrared region, high-cost efficiency, etc. To address those limitations, OSCs have been extended to non-fullerene acceptor materials.

The past decade has witnessed tremendous development in molecular design guidelines for the discovery and synthesis of novel quinoidal π-conjugated materials. End-capped quinoidal π-conjugated molecules are a subfamily, with π-extended core having two terminal groups. They possess high electron acceptability and low LUMO levels, which has made them excellent n-type semiconductor materials, with electron mobilities well in excess of 1 cm^2^ V^−1^ s^−1^. The nature of π-extended core and terminal group have a profound effect on the electrical and optoelectronic properties in those materials.

In more details below, we highlight selected examples from literature of the last three to four years on the design, synthesis, properties, and device performance of quinoidal π-conjugated materials, with special attention to emerging materials such as indophenine derivatives that show promising potential for high-performance devices. The reader is referred to several outstanding reviews that adress the use o quinoidal oligothiophene as n-channel materials in OFETs [8], the use of antiaromaticity and quinoid strategy as a tool for the design and synthesis of high-performance OFET materials [9], the role of the aromatic–quinoidal balance in determining the ground state of the quinoidal materials [10], the structural evolution of quinoidal conjugated polymers for employed for electronics application, with emphasis on the architecture of quinoidal frameworks and their attractive electronic structures [11]. 

In contrast, the present work investigates the recent improvements and design guidelines for quinoidal molecules with a chemical perspective. To our knowledge, this review represents one of the first compilation of the literature on the indophenine-based materials, covering their scope, progress, limitations, applications, and prospects. We also believe this review will benefit the organic electronics and photovoltaic communities by providing insights into the latest trends of new building blocks, which are available as high-performance materials. In the next section, we provide a review of state-of-the-art work in quinoidal semiconducting. Our focus will be on indophenine series, which has never been reviewed. We discuss the relationship between structural properties and energy levels in this family of molecules. 

## 3. Synthetic Tactics of π-Extended Quinoidal Acceptors

π-Conjugated quinoidal molecules are emerging materials for energy and optoelectronic applications. Two main strategies have been developed by chemists for their synthesis. These approaches lead to two different classes of quinoidal materials. The first approach involves embedding the quinoidal moiety into the core of an aromatic π-conjugation. The second approach, known as end-group functionalization, involves terminal capping of the terminal methylene sites by electron-withdrawing functional group (EWG). The functional groups, cyano, ester groups, or aryl groups, contribute to quinoidal character by blocking the reaction sites and delocalizing the spin. A π-extended core and terminal group have a profound effect on the electrical and optoelectronic performance of the resulting material. In the following section, the general approach of the synthesis of quinoidal families with various terminal units is briefly mentioned. Four structural modification tactics are discussed in detail, involving the introduction of the dicyanomethylene functionality at the terminal positions of a π-conjugated system, indandione-terminated and triphenylmethane π-conjugated quinoids, and finally indophenine family. Generally, there are two main synthesis routes for the preparation of end-capped π-conjugated quinoidal molecules according to the reactivity and the functionality of the end-group. For quinoidal dicyanomethylene-end capped molecules, the Takahashi reaction is the most efficient way for their synthesis. In this route, the dibrominated aromatic compounds allow for a Takahashi coupling and then for an oxidative dehydrogenation reaction to obtain the desired quinoidal forms (Route A, Figure 3). A quinoidal skeleton composed of four aryl groups bridged by a π-conjugated linker (Thiele’s hydrocarbon) is obtained via lithium–halogen exchange, followed by nucleophilic addition and reduction (route B). In the following section, the rationale behind the design of these molecules and the methodology developed for their synthesis will be discussed based on the different precursors of forming quinoidal forms. In addition to the above routes, new methods have been reported, such as intra- or inter-molecular radical–radical coupling reaction [12].

### 3.1. Recent Advances of Dicyanomethylene End-Capped π-Conjugated Small Molecules

Dicyanomethylene block is one of the moieties extensively involved in the construction of superior electron acceptor dyes. Its strong accepting ability arises from the vinyl extension of the conjugation length, which promotes the planarity of the molecule and tends to achieve materials with improved charge carrier mobilities. The first synthesis of TCNQ (Figure 4) was published in the 1960s [13]. Since then, various TCNQ derivatives have been reported [14]. This first synthesis was accomplished using a Knoevenagel condensation method, followed by oxidation using bromine [13]. To further improve the yield and simplify the reaction procedure, a new synthetic method has been developed by Takahashi and coworkers [15]. Since then, this reaction known as the Takahashi reaction has been widely used for preparing dicyanomethylene end-capped quinoidal molecules. In this reaction, the dihalogenated aromatics were converted into their respective quinoidal forms by a palladium-catalyzed coupling and followed by oxidation using Br_2_ or 2,3-dichloro-5,6-dicyano-1,4-benzoquinone (DDQ) [16]. A large number of review articles dedicated to TCNQ and its acceptor analogues are recommended from the point of view of design and synthesis [17], molecular architecture for optoelectronic applications [18], and optoelectronic devices.

By using this reaction, a range of quinoidal compounds have been synthesized, so we restrict the discussion to the most recent advances in TCNQ derivatives during the last four years (Figure 5, Figure 6, Figure 7 and Figure 8). Tao et al. presented the optimized route for the synthesis of quinoidal diketopyrrolopyrrole (DPP) derivative (DCM-DPP-C_13_) [19]. They found that the use of a large amount of sodium hydride damaged the DPP-C_13_-Br core, which is in sharp contrast with a previous study that used four equivalents of NaH [20]. Additionally, the reaction of the malononitrile anion with DPP-C_13_-Br over 0.5 equivalent amount of Pd catalyst distinguished from the typical Takahashi conditions, which merely require 1% of catalyst. The intermediate product can be further transformed into the resulting quinoid structure using the strong oxidant, i.e., saturated aqueous Br_2_ solution, accompanied by a small amount of bromination of the by-product at the C4 position of the thiophene ring.

Jiang and co-workers designed and synthetized S, N-heteroacenes quinoidal compounds (JH06–10, Figure 6) [21]. They have examined the structure–property relationship by investigating the effect of the central heterocyclic core’s length on the optoelectronic properties. By altering the quinone core structure, the key advantages are shown: Firstly, the N-alkyl substituent of pyrrole provides more solubility-enhancing groups; secondly, the number of species in this family of materials is greatly enriched; and finally, the alternating pyrrole and thiophene units and the electron-withdrawing terminal groups ensure a suitably low LUMO energy level of −4.22 eV.

Joseph and co-workers reported the synthesis of one type of thioalkyl-substituted bi- and ter-thiophene compounds (TSBTQ, DSTQ) with varying chain lengths for OFET applications (Figure 7) [22]. These molecules display a planar molecular structure with short intermolecular stacking distances. In addition, they possess a deep LUMO (<−4.0 eV) that inhibits air oxidation, resulting in excellent ambient stability. The deeper LUMO of −4.28 eV exhibited by DSTQ is due to a downward shift of 0.1 eV in the LUMO energy level caused by the introduction of the sulphur atom into the alkyl side of the molecule, compared to −4.18 eV for the alkylated analogue, with the HOMO energy levels of both also being somewhat affected. The replacement of the terthiophene spacer with bithiophene shows the same trend, and compound TSBTQ exhibits a low LUMO energy (−4.36 eV). These are in accordance with the previous work of Caneci et al., who concluded that the insertion of electron-donating group (EDG) onto π-conjugated bridge stabilized the quinoidal (closed-shell singlet) ground state to a great extent and increased the FMO energy levels [23]. 

The thioalkylthiophene is arranged in a face-to-face slipped π-π stacking arrangement, and the stacking distance of only 3.55 Å facilitates efficient charge transport. The shortest intermolecular distances for S⋯N and S⋯S of DSTQ-3 are 3.56 Å and 3.93 Å, respectively.

Longer π-quinoidal molecules possess singlet open-shell diradical character with one of the outstanding behaviors being the formation of the reversible diradical σ-dimer. Recently, Badia-Dominguez et al. [24] synthesized and characterized of ICz-CN with dicyanomethylene as the terminal unit at the 3,9 position, which is capable of forming σ-dimer configuration with two coplanar units with intriguing π-π interactions (Figure 8). They highlighted the properties of the open shell forms with respect to their σ-dimers (from ICz-CN to (ICz-CN)2). The reversibility of the monomer/σ-dimer transition was explored in the solid and solution states by varying the external stimulus (temperature or pressure), respectively. The chemistry of stable σ- and π-dimer of OSC radicals will be discussed in more detail in Section 3.7.

To stabilize the quinoidal electronic structure to counter the diradical structure, Yamamoto and coworkers designed and prepared BTQ-3 and BTQ-6, which have a benzo[c]thiophene core [25]. To avoid the problem of instability of the repeated thiophene framework, the reverse Diels-Alder reaction (220 °C and under reduced pressure) was used in the final step of the synthesis. The diiodide derivative intermediate was oxidized using DDQ following a Takahashi’s Pd-catalyzed coupling reaction. The electron accepting ability of these molecules was further strengthened by the insertion of a fluorine substituent at the β-position to the quinoidal skeleton. 

Thieno–isoindigo units have been employed extensively as receptor blocks for the OSC materials. The thiophene-based structure is more planar due to reduced intramolecular space resistance compared to the benzene ring, while the thiophene-thiophene linkage along the backbone maximizes the conjugation length and further enhances the intermolecular tight contacts. A series of dicyanomethylene end-capped quinoidal thieno–isoindigo with various alkyl chains (TIIQ) was synthesized by Facchetti group [26]. It was identified that the strategy of varying the branching points of the alkyl side chains resulted in varying the intermolecular stacking distances, which slightly tuned the optical and electrochemical properties of the resulting materials. These molecules are typically characterized by a very low LUMO energy level and show a mobility of up to 2.54 cm^2^ V^−1^ s^−1^ in OFET devices.

### 3.2. Indandione Terminated π-Conjugated Small Molecules

An alternative kind of quinolone moiety consists of indandione units capped are shown in Figure 9 [27]. These compounds feature low lie LUMO levels below −4.0 eV, and their LUMO levels are less affected by the central core π-bridge. The reduction in the band-gap of the material is mainly attributed to the rising HOMO levels as the π-conjugation increases. All these materials showed unipolar electron transport behaviors, exhibiting a maximum μ_e_ of 0.38 cm^2^ V^−1^ s^−1^ [28].

### 3.3. Analogs of Thiele’s and Chichibabin’s Dyes 

Among various classes of quinoidal OSC, Thiele (p-quinodimethane), and Chichibabin, polycyclic aromatic hydrocarbons have been extensively studied. They can be considered the first known quinoidal dyes. X-ray analysis of such molecules demonstrates a planar π-di-xylylene structure, which indicates a major contribution from the quinoidal resonance form in the crystal [29]. The length of all the bonds in the fundamental backbone lies in the range 1.371–1.448 Å, between the double and single bond values. In addition to this, the four benzene rings are rotated by an average of 43° rather than being aligned coplanarly. Given these structural findings, the improved kinetic stability of the Thiele hydrocarbons with respect to the parent may be largely due to steric effects. This feature is of considerable importance in the evaluation of design standards, synthesis, and purification of new analogues of Thiele and Chichibabin hydrocarbons. In synthesizing such quinoidal hydrocarbons, several modifications of initial procedures have been attempted. Some were successful; some were not. Initial studies involving 4,4′-Bis(diphenyl-hydroxymethyl)biphenyl required the use of n-butyllithium, and the resulting diol intermediate was subject to zinc debromination, but in very low yields. An alternative procedure employs the bis-carbenium salts, obtained by the protonation of the corresponding diols, which were successfully converted to the corresponding quinoidal compounds in good to reasonable yields (Figure 10) [30]. The stability of di-carbocations is an issue in the above synthetic approach. To overcome this shortcoming, Takeda T. et al. improved the product’s overall yield [31]. Thus, for the dehydration step, they changed the Brønsted acidic, HClO_4_, to a soft Lewis acid, TMSClO_4_, to generate the corresponding dicationic species. The corresponding target products were successfully obtained by treating the diols with TMSClO_4_ and then reducing with Zn. The yield of the dehydration step was raised sharply from 6% with HClO_4_ up to 99% with TMSClO_4_. The impact of terminal segments has been discussed, involving dibenzocycloheptatriene, fluoryl, and cyclopentadithiophene. Kobayashi et al. reported a variety of synthetic methods to prepare fluorenyl end-capped quinoidal systems. Three different approaches have been used for the generation of quinoidal derivatives, chemical, photochemical, and thermal [32].

Recently, an example of cyclopentadithiophene (CPDT) unit incorporated as an end group on the quinoidal structures allowing fine-tuning the FMO energy levels of the molecular components was reported by Wang and coworkers [33]. Even though CPDT is more donating than fluorenyl, the introduction of this group as a terminal part of quinoidal frameworks has noticeable stabilizing effects on the HOMO level, while the LUMO remains almost unaltered. The chemistry used to introduce this building block is similar to that of fluorenyl, based on the diol intermediates (Figure 11). The diols were synthesized by the double lithium–halogen exchange in di-halogenated aromatic and subsequent nucleophilic addition to aromatic ketone affording diols. The quinoidal structures were then attained in moderate yields through the reduction of the diol by SnCl2 in THF. We note that the diol intermediates directly afforded the corresponding quinoidal form, which greatly simplified the synthesis process. A similar protocol was used in the synthesis of Thiele and tetrabenzo-Chichibabin derivatives with terminal dibenzocycloheptatriene units (DBHept) [34].

### 3.4. Synthesis of Indophenine Dyes and Oxindole End-Capped Quinoidal Molecules

Adolph von Baeyer illustrated in 1879 that combining red isatin with benzole under sulfuric acid conditions generates the result of a deep blue compound, named indophenine [35]. Victor Meyer subsequently investigated thiophene in his study of this reaction and identified indophenine as the composition of isatin and thiophene [36]. The thiophene comes from the small amount of contaminants present in benzole, and the blue product is not composed of isatin with benzene as was initially thought. One century later, in 1993, Tormos et al. [37] suggested the presence of six stereoisomers of N-alkylated indophenine in solution by analysis of two-dimensional COSY Spectroscopy, as shown in Figure 12. Since these works, little interest has been dedicated to research in this area because the pre-existence of isomers affects the behaviors of molecular self-assembly, and highly crystalline films are in direct correlation with the mobility of the charge carriers.

The straightforward synthesis of indophenine molecules is acid-catalyzed condensation of isatin with thiophene (synthesis route A, Figure 13). Hwang and coworkers synthesized compounds **1a**–**1f** and **2a**–**2f**, whereby the quinoidal thiophene and selenophene were served as bridging groups, respectively [38]. They demonstrated that the different isomers own different characteristics in HOMO and LUMO energy levels. The trans-isomeric configuration between the thiophenes is energetically lower and thus more stable compared to the cis-isomer. Although the simplicity of the synthetic approach to indophenine, the formation of isomers leads to challenging purification of the final product and low yields of the reaction. In practice, the reaction results in two types of quinoidal molecules with dissimilar core lengths tending to be produced simultaneously, further decreasing the yield of the desired target product (compounds **1a**–**1f**, six isomers and compounds **1g**–**1i**, three isomers), in yield of ∼36% and ∼18%, respectively [39]. The quinoidal thiophene, regarded as the side-product, was separated and characterized using 2D NOESY NMR spectroscopy, which revealed that the main isomer showed an asymmetric *Z*, *E*-configuration [39]. Most recent investigations have shown that the dipole repulsion between the carbonyl groups in compound **6g** (*Z*, *E*) is the smallest of the three. Likewise, among the theoretical six geometric isomers, the quinone bithiophene with the (*Z*, *E*, *Z*) form is considered to be the major product [40]. Their dyeing behaviors, optical properties, and light stability were studied by Chen et al., who investigated the influence on halogen substitution (-F, -Cl), as well as the -SO_3_H (compounds **3**–**5**) in oxindole moiety on the photophysical properties and dyeing performance [41,42]. The location where the quinoidal skeleton occurs was studied by Bhanvadia et al. [43] They found that under similar experimental conditions, the product formed using a catalytic amount of concentrated H_2_SO_4_ was actually **6k** rather than **6j**. The quinoidal structure existing on the pyrroloindole dione unit was related to the steric bulk caused by the bromine substituents. One alternative approach for the synthesis of indophenine has been advanced by Ren et al. (route B) [44]. This reaction involves Sn^II^-mediated reductive aromatization of diols to afford the fully conjugated quinoidal products with subsequent oxidation of the intermediate aromatic intermediate. For instance, the aromatic ring containing the lithium substituent is added nucleophilically to the carbonyl group of isatin to obtain the diol intermediate, in a good yield, followed by reductive aromatization using SnCl_2_ reduction. Dehydrogenation using DDQ produced compound **7a**–**7c** as a mixture of three isomers. It should be noted that the reduction of the diols by SnCl_2_ can also lead directly to the quinoidal form, **7a**–**7c**, depending on the reaction conditions (route C) [45]. 

Three approaches have been used to overcome the severe issue of stereochemistry. For the first method, it relies on the steric hindrance obtained by the oxidation of thiophene, thereby reducing the amounts of isomers. Deng et al. found that the oxidation of thiophene unit in indophenine system provided a dual benefit: (i) a significant drop in both LUMO and HOMO; (ii) facilitated isomerization yielding a single isomeric product [46]. As shown in Figure 14, 3-chloroperoxybenzoic acid (*m*-CPBA, route D) was employed to oxidize a mixture of **8a**–**8c**, followed by refluxed in toluene to afford a pure single isomer of indophenine (compound **9c**). This is because **9c** is the most stable form among the three isomers. On the other hand, the isomerization process from **9b** or **9a** to **9c** is accelerated under mild heating conditions. Compared to compound **8**, the HOMO energy level of oxidization quinoidal compound **9** was significantly lowered to −5.91 eV (Table 1). Finally, the pure isomer was used as a co-monomer feed for constructing polymer (**P1**, Figure 15). **P1** exhibits the narrow band-gap, together with the low energy levels (LUMO: −3.98 eV & HOMO: −5.92 eV). **P2** and **P3** were synthesized later, and it was noteworthy that **P2** has a deeper LUMO of −4.09 eV compared to that of **P1**. The LUMO level was further reduced to −4.18 eV by introducing 2,2′-bithiazole (BTz) into the D–A polymer system (**P3**), which serves as the electron donor [47]. An impressive study of the impact of S, S-dioxided thiophene on the opto-physical and electrochemical properties of indophenine derivatives was reported by Hu and coworkers [48]. They synthesized several thiophene-S, S-dioxidized indophenine substituted at 5,5′ positions with electron donating/withdrawing groups (compound **10a**–**10e**). It was concluded that (i) oxidation of thiophene significantly improved the structural stability; (ii) insertion of S, S-dioxidized unit in quinoidal system caused a significant hypsochromic shift in the absorption; (iii) EDGs were beneficial in maintaining the quinoidal state, whereas EDGs had strong influence on the electron cloud density distribution. It was well known that the 5,5′ position was a cross-conjugated isoindigo, and thus the π-electrons were not well delocalized over the entire molecule, which dramatically affected the properties of the molecule. The impact on cross-conjugation of the substitution pattern on the electronic properties was investigated by Deng [49]. Compared to unsubstituted **11a**, the substitution at 5- or 6- position lowers the FMO energy levels; however, the position of substitution has little effect. The DFT calculations revealed that the substitution at 6,6′ position (compound **11b**) participates in LUMO and HOMO conjugation paths by inductive resonance effects, while 5,5′-dibromo substituents, compound **10c**, only participate in the HOMO wave function through the inductive and resonance effects. The charge transport properties are found to be less impacted by the position of the substitution. The electron mobility of the molecule substituted at position 5 rising up to 0.071 cm^2^ V^−1^ s^−1^, a value comparable to that of compound **11b**, 0.11 cm^2^ V^−1^ s^−1^, although the compound **11b** has a more extended π-conjugated system. Geng’s group further expanded the range of products obtained by relying on this type of synthetic strategy (route D). Oxidation of the mixture **7a**–**7c** with *m*-CPBA, followed by thermal isomerization (120 °C in toluene), gives a single-isomer (compound **12c**) [44]. Bromine functionalized isatin derivatives can be further used to design more sophisticated macromolecular architectures, and three different donors were introduced into the synthesis of three polymers, **P4**–**P6** (Figure 15). The LUMO level of these polymers is constant at −4.04 eV. On the one hand, they show a variation in HOMO from −6.00 to −5.91 eV because of the different donor abilities of donor-acceptor (Table 2). Upon substitution of peripheral hydrogen atoms with fluorine in the terminal isatin units, the resulting fluorinated indophenine dyes exhibited lower LUMO/HOMO levels [50]. Compound **13a**–**13c** with varying amounts of fluorine substituents on the oxindole have been reported. All derivatives exhibit n-type transport behavior, where the electron mobility is correlated with the number of fluorine atoms. Compound **13b** exhibited the highest electron mobility of 0.16 cm^2^ V^−1^ s^−1^, which is associated with its two-dimensional electron transport mode and highly ordered film.

The second method employed to drive the reaction to single isomer formation involves the utilization of non-covalent conformational locking and steric repulsion as the driving forces. The original synthesis of indophenine involved the condensation of thiophene and isatin under concentrated sulfuric acid. Nevertheless, this condition resulted in a complex mixture as reported by Cava et al. [37]. This limitation was elegantly overcome by Ren and coworkers, who proposed mild reaction conditions using Sn^II^-mediated reductive aromatization of diol to afford the fully quinoidal form (compound **14**) without oxidation of the intermediate aromatic form [45]. For the final step, air oxidation led to the formation of targeted compounds in good yield, although the authors did not mention the role of oxygen in their discussion. The intramolecular O⋯H interactions present in the compound **14** and confirm a well-defined conformation, as shown in the Figure 14, with no other isomers generated. The thieno[3,4-b] thiophene favors the stable *E* conformation, and its large π surface and short intermolecular S⋯S interactions also favour hole transport. p-type semiconductors compound **14** exhibited excellent unipolar hole mobility up to 0.15 cm^2^ V^−1^ s^−1^. Geng’s group further proposed the chlorination strategy for the conformation of the locked indophenine derivatives [51]. The energy gap among **7a**–**7c** was determined by DFT calculations to be less than 1 kcal/mol, and this variation was significantly enhanced by the insertion of a chlorine atom. By using non-bonded covalent interaction analysis, it was shown that although there are certain S⋯H and O⋯Cl interactions, the Cl⋯H and S⋯O interactions tend to be stronger. The maximum absorption peaks of 15b and 15c were red-shifted compared to 15a, and the introduction of F and Br reduced E_HOMO_/E_LUMO_ significantly (Table 1).

Further development of the selective synthesis of a single isomer was accomplished by Pappenfus et al., who succeeded in producing a single isomer using the noncovalent conformational lock approach. Route E outlines the reaction of N-alkylisatin with 3,4-propylenedioxythiophenes (ProDOT) in toluene catalyzed by sulfuric acid to form quinoidal molecular (compound **16**). The intrarmolecular interaction was the driving force for obtaining one of the isomers [52]. Using a similar strategy, O⋯S non-covalent interactions and steric repulsion were employed towards the synthesis of bis-QEDOT, compound **17** [53]. Similarly, Kim et al. found that compound **17** and **18** were synthesized simultaneously by the indophenine reaction under sulfuric acid catalytic conditions with yields of 40% and 12%, respectively [54]. Both quinoid monomers were confirmed to have a single geometric structure with NMR analysis, which also confirms the intramolecular nonbonding S···O interactions. **P7** and **P8** have been obtained by polymerization reaction using mono- and di-EDOT with vinyl. The substitution of O atoms in the quinoidal core by S atoms in compound **19** improve π-electron accepting ability of the system [55]. The S-based analogs exhibit lower FMO levels with narrowed band-gaps. However, the authors noted that compound **19** has two isomers, meaning that it can be isomerized under ambient conditions due to the low activation energy of quinoidal molecules. The corresponding monomers were polymerized with bithiophene to obtain polymers **P9** and **P10**, respectively. The Grazing incidence wide-angle X-ray scattering results showed **P9** film with ordered crystallization and better hole mobility in the OFET. Most recently, the substitution of the H with methoxy group on thiophene was promoted in the O⋯S conformational locking of indophenine [56]. In addition, three polymers consisting of bis-QEDOT (**P11**−**P13**, Figure 15) were synthesized by a Stille polymerization reaction. The high HOMO energy level of these polymers, close to −4.5 eV, makes them promising p-type OFET materials. After substitution of methoxy at the 3,4 position of thiophene, the reaction gave only a single *Z*, *E*, *Z* conformation of product 20. The X-ray structure shows a highly planar core with a distance of 2.69 Å between the S and O atoms, indicating the involvement of spatially non-covalent interactions, which leads to a dihedral angle of almost 0° between the two thiophene molecular units.

Recently, we have developed the idea of a novel variant of the indophenine condition [57]. The previously reported route involved a one-pot procedural process starting with isatin and a suitable 5-membered aromatic heterocycle. However, as we have discussed above, the yield under this condition was low, and the selectivity was poor. We propose to modify the standard conditions to improve the yields by using tertiary alcohols as starting materials (Compound **21a–c** in Route F). This strategy proved to be particularly selective, isomer-free for the synthesis, avoiding any complex purification problems, all of which resulted in the product yield being improved. A range of quinoidal materials with variable termini and modified conjugate cores have been synthesized and characterized. 

### 3.5. Molecular and Electronic Structures of Oxindole Terminated Quinoidal Molecules: Electrochemical and Optical Properties

Indophenine dye is an electron-deficient building block that can be regarded as an isoindigo analogue with a central π-core extension. It contains two electron-withdrawing carbonyl units of oxindole moiety to gain better stability for the π-extended quinoidal structures. In addition, it provides a deep LUMO level (−4.0 eV). The indophenine structure after DFT optimization indicates an nearly planar skeletal configuration [41]. The electron cloud density is mostly delocalized over the quinoidal π-system. The FMO levels are sensitive to the variation of the substituents. The general trend is that an EDG can increase the energy level, while the EWG plays the opposite role. For instance, by introducing a halogen atom (fluorine or chlorine) on benzene of oxindole ring, the energy level of the material is correspondingly and significantly reduced. Another promising method for tuning the electronic energy levels of indophenine molecules is to alter the quinoidal π-conjugated core. To illustrate this, Hu and coworkers assessed the energy levels in a series of substituted indophenine compounds where the sulfur atom of thiophene is oxidized to the S, S-dioxided thiophene. From DFT calculation and electrochemical study, a shift of 0.5 eV of the LUMO levels is observed, which clearly demonstrated the strong electron affinity of compound **12c**. Replacing of the thiophene in indophenine molecule with ProDOT or EDOT resulted in materials with similar LUMO levels, as shown in Table 1. The variation in HOMO in compound **16**/**17** can be explained by the strength of the electron donating capacity of ProDOT/EDOT vs. thiophene group.

### 3.6. Crystal Packing of Indophenine Molecules with O⋯S Conformational Locks

In OSCs, the non-covalent conformational lock has been employed as a practical means of improving the planarity of the skeleton, enhancing molecular stacking and improving the mobility of charge carriers. Components bearing O⋯S non-covalent interactions are already extensively employed as conformational locks to construct OSCs, and there are many examples of EDOT-based materials with an O⋯S bond locked conformation. For instance, bi-EDOT’s crystal structure indicates intense intramolecular non-covalent interactions **[58]**. Analysis of its crystal structure allows calculation of a distancing of 2.92 Å between the O and S inside the molecule, which is smaller compared to the sum of the van der Waals radii of the two atoms. This force locks the conjugated structure into an almost planar conformation with a torsion angle of 6.9° (Figure 16a). Another system where intramolecular O⋯S forces are also present is based on the dicarboxylic-bithiophene moiety. Figure 16b illustrates the intense forces that exist within the molecule due to the close spatial distance of the S⋯O, resulting in a near-flat skeletal structure with a minor torsion angle of 2.7°. Likewise, the double ProDOT core in compound **16** favors a macroplanar structure throughout the entire molecule, with an infinitely sliding layer-by-molecule arrangement of an interplanar distance of just 3.57 Å (Figure 16c). Figure 16d illustrates that the distances between the S atom and the two adjacent O atoms within the molecule are 2.670 and 2.788 Å, respectively, and that those contacts are obviously shorter than the sum of the van der Waals radii of S and O (3.25 Å). 

It is interesting is that the controlled generation of isomerization to the single isomer 9c is promoted in compounds **9a**–**9c** due to the introduction of steric hindrance. Figure 16e analysis further confirms that the isomers adopt the *E*, *E*, *E* conformation and exhibit a completely planar skeleton. The introduction of two additional O atoms in each thiophene presents an almost perpendicular angle to the entire plane. These molecules form a sliding face-to-face stacking pattern with an interlayer distance of 3.76 Å. Intermolecular hydrogen bonds have been formed between the H_4_ proton of the thiophene and the O atom of the oxindole unit, measuring the H to O distance and the angle of this bond at 2.457 Å and 128.58°, respectively (Figure 16f). By replacing the π-bridge of indophenine dye, bis-EDOT unit (compound **17**), with tetra-methoxy-bithiophene (compound **20a**), a single regio-isomer was obtained, via the manipulation of the configuration via a joint effect of steric hindrance and intermolecular contact to lock the conformation. [56] Thus, according to Figure 16g, compound **20a** molecule exhibits an almost planar structure and maintains the *Z*, *E*, *Z* conformation, with a torsion angle of only 6.7° between isatin and DMOT. The spatial distances between the two O atoms and the central S atom are found to be 2.84 Å and 2.69 Å, respectively. As previously discussed, the contact of S⋯O highlights the role of intramolecular non-covalent interactions in the conformational lock. The role of intramolecular interactions to induce conformational lock was further demonstrated in di-chlorinated bithiophene bridge in indophenine (compounds **15a–c**) [51]. The presence of intramolecular Cl⋯H and S⋯O non-covalent interactions in Figure 16h contributes to the stabilization of the *Z*, *Z* conformation.

### 3.7. Chemical Stability of Diradicals Based Quinoidal Molecules

Although quinoidal molecules have emerged as excellent ambipolar and n-channel OFET materials, only little literature exists on their stability. In fact, such materials are commonly thought to present two types of resonance structures: one quinoidal and the other diradical. Diradical structures contain at least one unpaired electron, which renders them more reactive and sensitive species, and they are so facing stability issues [59]. Few studies from different groups: Haley [60], Wu [61], Chi [62], Casado [63], Tobe [64], and Navarette [65] have shed some light on the stability of diradical hydrocarbons, providing guidelines on how to design stable diradical π-conjugated materials. The major challenge in designing these materials is to achieve a balance between their stability and targeted properties. Different strategies to stabilize π-radicals are used [66]. Firstly, the kinetic stabilization relies on introducing a bulky substituent in the close vicinity of the radical center to hinder its dimerization. Secondly, according to spin delocalization approach, π-radicals with more spin delocalization show higher stable because the spin density is diluted, causing diminished reactivities. Finally, the introduction of EDGs provides thermodynamic stabilization through their conjugative and inductive effects. Dimerization is considered the main way by which radicals are degraded. This can be categorized as σ- and π-dimerization. The former involves the formation of a covalent (σ) bond between two radicals. The latter, often observed in the association of π-conjugated radicals, consists of the formation of a stacked pair of radicals via π-orbitals. Some dicyanomethylene end-capped oligothiophenes having a low number of ring possess a closed-shell quinonidal structure; while as the length of the oligomer increases, its ground state is transformed into an aromatic open-shell diradical. These open-shell structures can reversibly dimerize when stimulated by external concentration, temperature, or pressure conditions. In a recent study, Zafra and coworkers demonstrated that quinoidal is capable of forming σ- or π-dimers according to the properties of the terminal moieties. They found that the different conformations of the dimer, such as open, extended, and completely closed, were caused by different mechanisms [67] (Figure 17).

A fundamental understanding of how a π-conjugated core impacts the reactivity of σ-bonds was reported by Badía-Domínguez et al. [24]. Diradical ICz-CN forms two long-range σ-bonds between the dicyanomethylene substituents during complete dimerization to (ICz-CN)_2_, as shown in Figure 8. The reversibility of this transformation has been discussed in solid and solution state. Butyl-substituted phenalenyl-based neutral radical materials assume great importance in terms of potential applications. The understanding of their chemistry, as well as the design rule for synthetizing stable open-shell phenalenyl structures have been reviewed [68]. For instance, fusing an anthracene with bisphenalenyl units generates a relatively stable Kekulé molecule with a very significant singlet diradical character [69]. 

Kubo and his colleagues investigated the correlation between three modes of cyclization and the stability of radicals. They prepared three radicals of fluorenyl fused naphthalene rings, fluorenyl-naphthalene-1, -2, -3, that differ in their mode of annelation [70]. The half-lifetime of the materials for the three models are 7 days, 3.5 days, and 43 days, respectively. The variation in stability is related to the kinetic stability as well as the thermodynamic stability achieved through the spin-off domain. Bis-imidazolyl radical moiety was recently explored by Abe group as a novel photochromic molecular system [71]. It was revealed a difference in the stability of the singlet and triplet state, and upon excitation at 609 nm, the imidazolyl core dimerizes to the colorless dimer (Figure 18).

In this section, we analyzed OSC radicals showing reversible association-dissociation behavior, forming σ- and π-dimer, from the viewpoint of molecular design. The recent studies represent significant advances in understanding the formation of highly stabilized intramolecular dimer radical, σ- or π-dimer, by the reversible monomer–dimer transition in the solid state. All these results pave the way to future directions on how diradical character can be controlled or modulated with external stimuli.

## 4. Device Applications

### 4.1. Quinoidal-Based Materials for OFETs

Since the seminal works of Frisbie et al. on the use of dicyanomethylene end-capped quinoidal oligothiophenes as the active material in n-type OFET, various quinoidal materials have been investigated [72,73]. In the last two decades, an impressive range of OSCs (n-type small molecules or polymers) have been reported and shown to have high electron mobility (0.5–1.0 cm^2^ V^−1^ s^−1^). The chemical modification approach has shown to be an effective strategy for synthetizing efficient stable n-type OFETs [74]. 

#### 4.1.1. OFET-Materials Based on Dicyanomethylene End-Capped Quinoidal Molecules (n-Type and Ambipolar)

Dicyanomethylene end-capped thiophene-based quinoidal compounds are an outstanding family of n-type OSC due to the existence of strong EWG at the end of quinoidal structure, thus affording a low-lying LUMO level. The central quinoidal cores promote π-stacking, thus inducing strong non-covalent intermolecular interactions, which may lead to increase the charge carrier mobility. Thus, several groups have explored the use of dicyanomethylene quinoidal molecules as n-type OSCs for OFET application (Figure 19). TIIQ-b16 affords OFETs with good mobility (μ_e_ = 2.54 cm^2^ V^−1^ s^−1^) [26]. In this molecular structure, the quinoidal structure results in a low-lying LUMO energy (−4.16 eV). Charge transport of these materials was investigated along with their morphological and microstructural studies. The fused planar aromatic structure having a five-membered ring, which is known to exhibit anti-aromaticity character, was combined with a strong EWG at the molecular termini to further lower the LUMO level. As a representative example, dithiarubicene is an analog of rubicene with a high electron affinity. The effect of multiply cyano substituents (BisDCNE, BisTCNE, and TCNQE) on OFET behavior was studied by Tsukamoto et al. [75]. The LUMO/HOMO energy levels measured with cyclic voltammetry are −3.97/−5.69 eV for BisDCNE, −4.23/−5.99 eV for BisTCNE, and −4.20/−5.62 eV for TCNQE. BisTCNE exhibits both a deeper LUMO and HOMO energy levels than TCNQE. Incorporation of three cyano units at the end of dithiarubicene core led to a slightly decrease in both LUMO and HOMO of BisTCNE owing to the electron-withdrawing capacity of the cyano. OFET devices based on BisTCNE exhibit a better performance than those based on TCNQE. The optimum mobility of 0.055 cm^2^ V^−1^ s^−1^ was demonstrated for BisTCNE. Ren and coworkers designed and synthetized a new molecule QDPPBTT featuring extremely low LUMO levels (−4.37 eV) [76]. DPP was incorporated without alkyl chain substitution to enhance the hydrogen bonding of DPP and better crystallinity. QDPPBTT-based OFET materials show promising electron mobility (0.13 cm^2^ V^–1^ s^–1^).

Incorporation of fluorine atoms into the quinoid system resulted in FTQ1, FTQ2, and FTQ-3 (Figure 19) [77]. It was found that the HOMO and LUMO levels for fluorinated molecules were much lower than those of non-fluorinated analogues. When we compare fluorinated versus non-fluorinated analogues, a down shift of the LUMO position by 0.2 eV was observed. The fluorinated terthiophene FTQ-2 adopts slipped π-π stacking. Furthermore, the fluorine substitution locks the planar conformation through the non-covalent bonding interaction of F and S. FTQ-3 annealed at 130 °C exhibited the highest mobility among the three. Quinoidal oligothiophenes (QBDT and QTBDT-3H) have been developed by Lin et al. [78]. The LUMO levels of the two molecules are very close to each other. Increasing the conjugation length by adding one thiophene ring from QBDT to QTBDT-3H enhances the diradical character. Spin-coated films of QTBDT-3H and QBDT in OFET devices exhibited ambipolar transport behavior for QTBDT-3H and unipolar transport property for QBDT. 

#### 4.1.2. OFET Performance of Polymers Developed as n-Type Channel Materials

Thieno-quinoidal systems with oxindole group as the end group have proven to be excellent electron-deficient building blocks. Additionally, the outer benzene ring of the oxindole offers an excellent opportunity to further extend the conjugation length, via introduction of bromine atom and subsequent polymerization with traditional palladium-catalyzed polycondensation. Deng and co-workers first reported the n-type quinoidal polymers (**P1**, Figure 15). These materials exhibited a moderate electron mobility (Table 3). Later, they reported the synthesis of two other polymers (**P2**, **P3**), with lower LUMO energy levels, suggesting that they have a lower barrier to electron injection in OFET. The electron mobility μ_e_ was further enhanced for OFET device due to more ordered films of **P2**. However, **P3** showed much lower performance due to its amorphous film and poorer crystallinity. Replacing quinoidal bithiophene with quinoidal thienothiophene yielded polymer **P6**, with mobility of 0.45 cm^2^ V^−1^ s^−1^ and air stability in OFETs, while **P4** performed the worst, with maximum mobility of merely 0.004 cm^2^ V^−1^ s^−1^ [44].

N-type OSC-materials are less developed than their p-type counterpart, due to their instability in air as well as their low electron mobility. Therefore, design of n-type OSC materials with high electron mobility is an emerging field. However, high-performance OFETs device relies on not only the properties of n-type OSC active layer but also the dielectrics and device processing techniques. An ideal n-type OSC semiconductors with high electron mobility and good stability in air should have a planar molecular structure with short intermolecular stacking distances. In addition, they should possess a deep LUMO (<−4.0 eV) to inhibit air oxidation, thus resulting in excellent ambient stability.

#### 4.1.3. OFET Devices Performance of Polymers Used in Transistor Applications (p-Type and Ambipolar)

**P11**−**P13** were synthesized using the Stille polymerization reaction (Figure 15). Among them, **P11** exhibits the highest hole mobility under optimized conditions of processing with an annealing temperature of 300 °C (Table 4). Nevertheless, the performance is still relatively low compared to that of quinoidal materials. The insertion of fluorine atoms of oxindole ring in **P16** has good impact on the coplanarity of the polymer chains with the best mobility up to 2.70 cm^2^ V^−1^ s^−1^. Huang et al. investigated P18 that exhibits a reversible structure between aromatic, open-shell form and quinoidal, closed-shell form [79]. Hence, the energy level of LUMO of P18 has been lowered by 0.27 eV when compared to that of isoindigo analogue. The best OFET performance was achieved when the thin film was annealed at 300 °C. Using similar design strategy, Kim et al. recently developed novel conjugated polymers **P19** and **P20** by incorporating quinoid moieties with different conjugation lengths units. These copolymers exhibit close- and open-shell biradical character depending on their quinoidal moiety. The increased conjugation length results in the increased properties of diradical character and a open-shell structure in both monomer and the resulting copolymer. On the other hand, a smaller conjugation length of the quinoidal core maintains a closed-shell quinoid structure, and the resulting copolymer exhibits a high backbone coplanarity and a strong intermolecular interaction. These characteristics are beneficial for charge transport. The optimum of OFET device performance for **P21** and **P22** occurred after annealing at 250 °C with a hole and an electron mobilities of 4.82 and 1.11 cm^2^ V^−1^ s^−1^ and 8.09 and 0.74 cm^2^ V^−1^ s^−1^, respectively. Hwang and coworkers synthesized quinoidal copolymer (**P23**), which has a quinoidal indophenine unit linked with a vinylene unit [80]. Introducing a vinylene link between the two indophenine units induces a highly coplanar. Ambipolar charge transport behaviors were identified in OFET devices. The HOMO and LUMO energy were found to be −5.08 and −3.79 eV, respectively. A further enhancement of the open-shell character of the conjugated copolymer with ambipolar semiconducting behavior was successfully achieved [81]. **P24** exhibits an ambipolar charge-transport behavior in OFET devices (Table 4); however, the six isomers could not be successfully isolated [81].

In summary, this section highlights different synthetic strategies employed, in recent years, to develop novel quinoidal molecules and provides an overview of their use as a useful building block to produce novel polymers with high-spin characteristics, tunable optoelectronic properties, as well as ambipolar and n-type semiconducting property.

### 4.2. Organic Diradical TE Materials

As noted during the introduction, the developments of TE-materials are very attractive from the point of view of achieving more efficient devices for energy harvesting. Both p-type and n-type materials are demanded in the advancement of the TE field. PEDOT:PSS is a benchmark p-type conducting polymer for TE applications, with a promising power factor of 47 μW m^−1^ K^−2^ and higher electrical conductivity of 900 S cm^−1^ [82]. In contrast, very few n-type polymer TE materials exhibited a moderate TE property. For instance, few reported materials can exhibit a conductivity higher than 90 S cm^−1^ for N doped conjugated polymer-based thiophene-fused benzodifurandione and good power factor of 106 mW m^−1^ K^−2^ for doped copolymer [83]. There are many recent reviews covering the materials design [8,84] and n-type doping techniques [85]. 

The reason why n-type TE materials are less developed lies in the lack of materials with deeper LUMO levels for efficient electron injection, stable charge transport, and an efficient n-type-doping process. N-doping of OSCs is extremely challenging because of the lack of OSC with deep LUMO (−4.7 eV) to effectively stabilize n-doping under ambient conditions. Recently, many studies have explored n-doping OSCs using various organic and inorganic salts and pointed out the air-stability issue of doped materials. For instance, Katz et al. reported air-stable n-doped ClBDPPV with low LUMO energy of −4.3 eV (Figure 20). The electrical conductivity dropped by 50% within 24 h [86]. To overcome this limitation and generate materials with improved stability, diradicaloid materials with a deeper LUMO energy represent an attractive approach. Yuan and coworkers found that diradical character and deep LUMO are favorable for stable and excellent thermoelectric performance [87]. 2DQTT provided the best compromise between stability and enhanced electrical performance. The substitution of Se for S is of assistance in changing the electronic properties of the system, due to the strong intermolecular Se⋯Se interactions and high polarizability of Se [88]. 

### 4.3. Quinoidal Semiconducting Materials for Photovoltaic Applications

Quinoidal π-conjugated materials show outstanding optoelectronic properties in terms of a low band-gap with high absorption coefficients and remarkable charge-transport properties. Despite these features, they are less investigated in photovoltaic applications. In contrast to aromatic forms, the optical band-gap of quinoidal form materials allows readily adjusting to obtain near-infrared light. Nonetheless, the main challenge in achieving efficient organic solar cells is obtaining high crystallinity. This prevents their use in photovoltaic devices, and there are only few reports on this filed. For instance, Ren et al. designed dithienoindophenine derivatives (μ_h_, 0.22 cm^2^ V^−1^ s^−1^) with suitable HOMO and LUMO (DTIP-I and DTIP-o in Figure 21). DTIP-o exhibits a better PCE of 4.07% [45]. Another application of quinoidal materials in photovoltaic is their use as non-fullerene materials. Materials with fused forms and with strong EWGs are promising acceptor materials for organic photovoltaics. Indacenodithiophene-based small molecular acceptor (ITIC) is an archetype n-type OSC and possesses a broad and strong absorption spectrum. This non-fullerene acceptor has been used in device reaching up to 12% in PCE [7]. Replacing the phenyl ring of the dicyanomethylene–indanone moiety with thiophene ring enhances the quinoidal character, which reduces the optical band-gap and enhances the near-IR absorptivity [89]. The maximum PCE and average PCE for ITCT were 11.27% and 10.99%, respectively.

## 5. Conclusions

In this review, we summarized the recent state-of-the-art progress and some guideline for the design of quinoidal organic semiconducting materials. Over the past decade, high-performance n-type and ambipolar conjugated quinoidal systems have been extensively investigated. Indophenine-based structures showed a remarkably high electron and hole mobilities. Different synthetic strategies to improve the yield, reaction scale, regioselectivity, and product functionality of indophenine dyes have been discussed. The indophenine reaction simultaneously produces quinoidal compounds of different bridging lengths and contains multiple isomers, which greatly limits the yield and purity of the material. A major problem facing is the indophenine reaction produces a mixture of isomers, which results in an intricate and complex NMR spectrum and making it difficult in analyzing and quantifying proportions of each isomer. The presence of isomers is correlated with the performance of the device. We believe that the separation of materials with only one and two bridging groups is facilitated by exploiting the difference in solubility of small molecular materials of different lengths in various organic solvents. By engineering the π- bridge group, the intermolecular interactions can be used to drive the formation of one isomer. Few materials based on indophenine showed good performance in terms of hole mobility up to 8.09 cm^2^ V^−1^ s^−1^, but most materials exhibit a mobility of 1 cm^2^ V^−1^ s^−1^. The electron mobility was shown to be worse than the hole mobility, which is related to the electron donating nature of the thiophene or EDOT rings contained in their structures. Designing unipolar high electron mobility materials or ambipolar materials with balanced ambipolar properties remains a challenge for organic chemists. Quinoidal structures are among the emerging building blocks for constructing high-performance n-type and ambipolar semiconducting materials. Rational structural design strategies are an efficient approach to enhance their electrical and optical properties. For instance, the introduction of strong electron-withdrawing groups, i.e., fluorine and chlorine atoms, into the oxindole ring can enhance these properties without affecting the selectivity of the reaction. However, single isomer formation enhances the planarity of the material, while significantly lowering the LUMO energy level of the material, and subsequently enhance its electron transport ability and stability in air. Another possible direction in this field comes from the processing of devices, which often require the use of chlorinated solvents such as chloroform and chlorobenzene due to the poor solubility of indophenine-based materials. This increases production costs and environmental pollution. Therefore, engineering the side chains in the materials, for instance, the introduction of more soluble chains on oxindole ring facilitates the preparation of intrinsically highly soluble materials. This modification makes the quinoloidal molecules more soluble in non-halogenated organic solvents, which in turn facilitates the fabrication of devices via solution processes. On the other hand, most of the reported polymeric materials based on indophenine use Stille coupling reactions, which involves organotin intermediates. Thus, this could lead to potential environmental hazards. We believe that the use of C-H activated cross-coupling and metal-free polymerization routes to π-conjugated polymers will inspire new research directions in this field. The studies of different molecular structures of conjugated quinoidal materials, with variable termini and π-conjugated core, are helpful to understand the structure-property relationships. The effects of various end-groups, π-conjugated core, conjugated backbones, and side chains on the OFET performance are able to provide guides for synthesizing new generations of quinoidal or diradical materials with tunable optoelectronic properties and more outstanding charge carrier mobility of up 8.09 cm^2^ V^−1^ s^−1^ in OFET devices.

## Figures and Tables

**Figure 1 materials-16-02474-f001:**
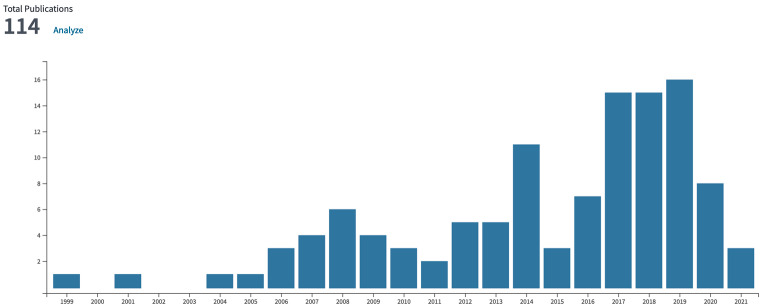
The number of reviews on the topic n-type OSCs from Thomson Reuters Web of Science is 114, of which the number of Quinoidal OSCs reviews is 9. Of the 106 indophenine references, no review articles have been reported according to Scifinder.

**Figure 2 materials-16-02474-f002:**
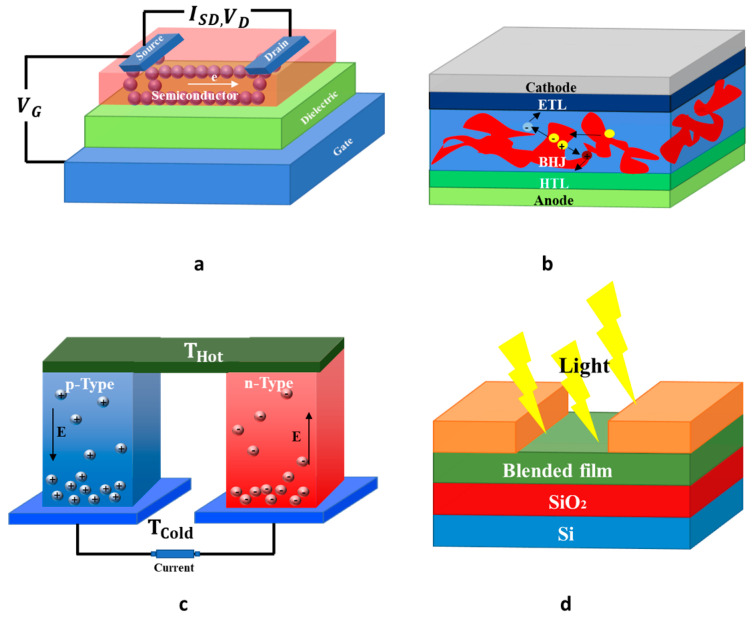
Representative thin-film optoelectronic devices that use OSC including (**a**) a bottom-gate top-contact OFET; (**b**) a bulk heterojunction organic solar cell; (**c**) an organic TE device; (**d**) a photo transistor.

**Figure 3 materials-16-02474-f003:**
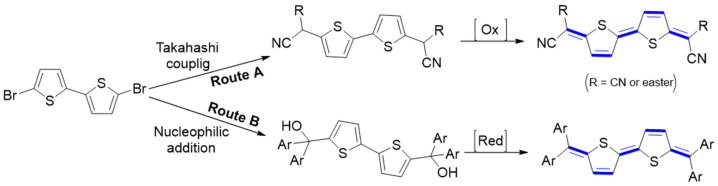
Typical synthetic strategies for the preparation of quinoidal π-conjugated molecules.

**Figure 4 materials-16-02474-f004:**
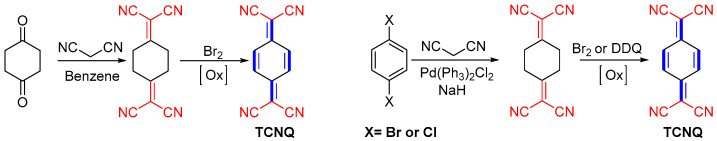
Synthetic route to TCNQ. Left: the original TCNQ synthesis. Right: the Takahashi’s synthesis.

**Figure 5 materials-16-02474-f005:**
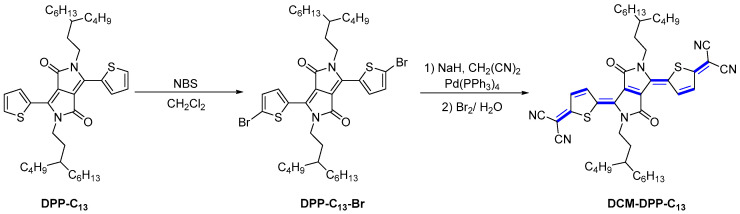
The synthetic route to dicyanomethylene end-capped DPP derivatives.

**Figure 6 materials-16-02474-f006:**
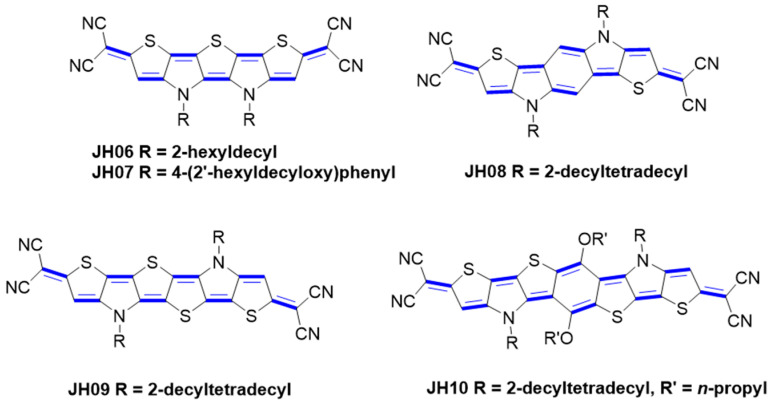
Molecular structures of dicyanomethylene end-capped quinoidal S, N-heteroacenes derivatives.

**Figure 7 materials-16-02474-f007:**
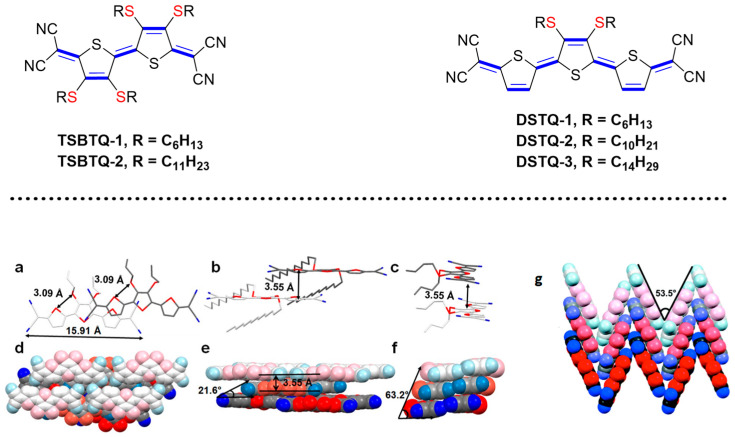
Top: Molecular structures of bithiophene and terthiophene-based small molecule quinoids TSBTQ and DSTQ; Bottom: Representative molecular packing modes of thiolakylated quinoidal oligothiophene: terthiophene and bithiophene (**a**–**c**) Top view, front view, and side view of DSTQ-3; (**d**–**f**) Molecular packing arrangement of DSTQ 1–3; (**g**) A herringbone mode of TSBTQ-1. Adapted with permission from reference [22]. Copyright (2020), *J. Mater. Chem. C*.

**Figure 8 materials-16-02474-f008:**
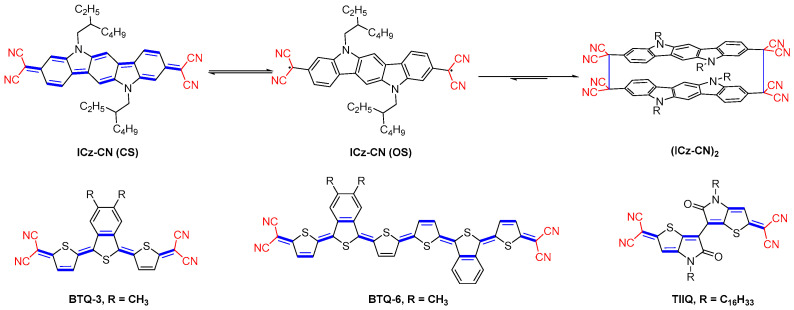
Molecular structures of ICz-CN (top), BTQ-3, BTQ-6, and TIIQ.

**Figure 9 materials-16-02474-f009:**
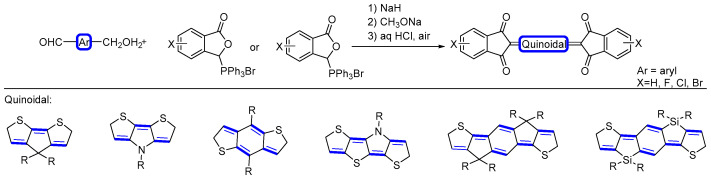
The synthetic route to construct indandione-terminated quinoidal materials.

**Figure 10 materials-16-02474-f010:**
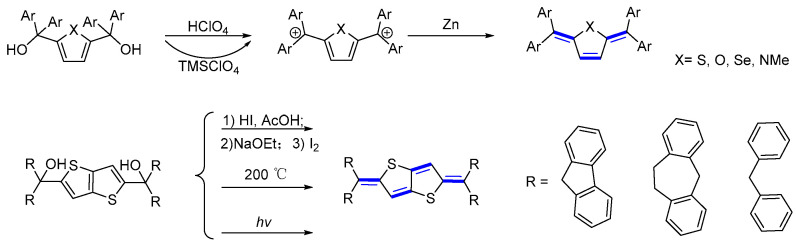
The synthetic route to Thiele’s and Chichibabin’s hydrocarbons.

**Figure 11 materials-16-02474-f011:**
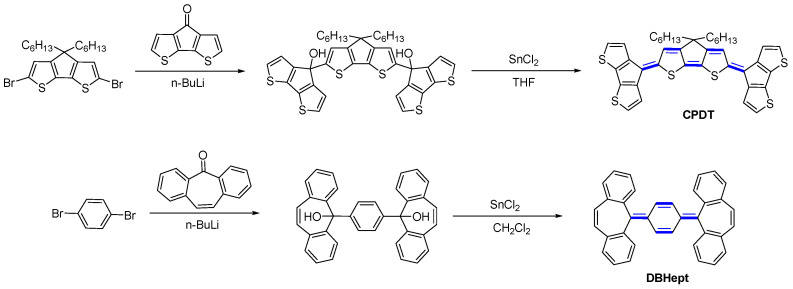
The synthetic route to CPDT and DBHept quinoids.

**Figure 12 materials-16-02474-f012:**
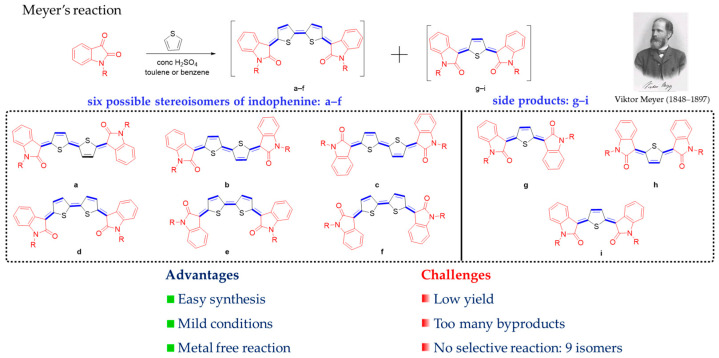
The classical method to the preparation of indophenine dyes.

**Figure 13 materials-16-02474-f013:**
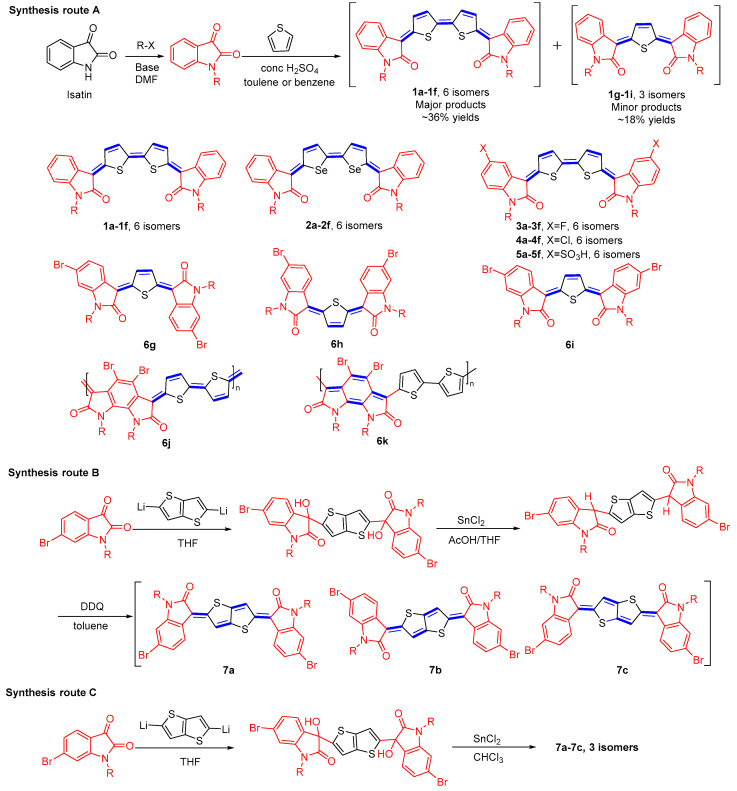
Different synthesis routes for the preparation of indophenine and their chemical structures of indophenine dyes isomers reported in the literature.

**Figure 14 materials-16-02474-f014:**
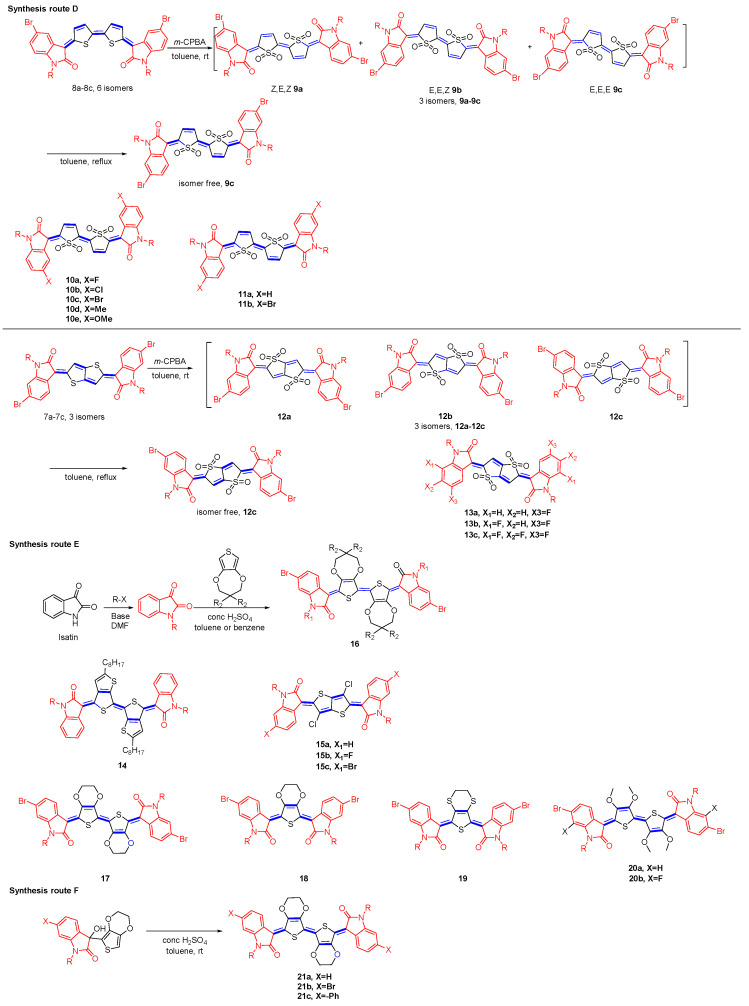
Different synthesis routes for the preparation of isomerically pure indophenine dyes and their chemical structures reported in the literature.

**Figure 15 materials-16-02474-f015:**
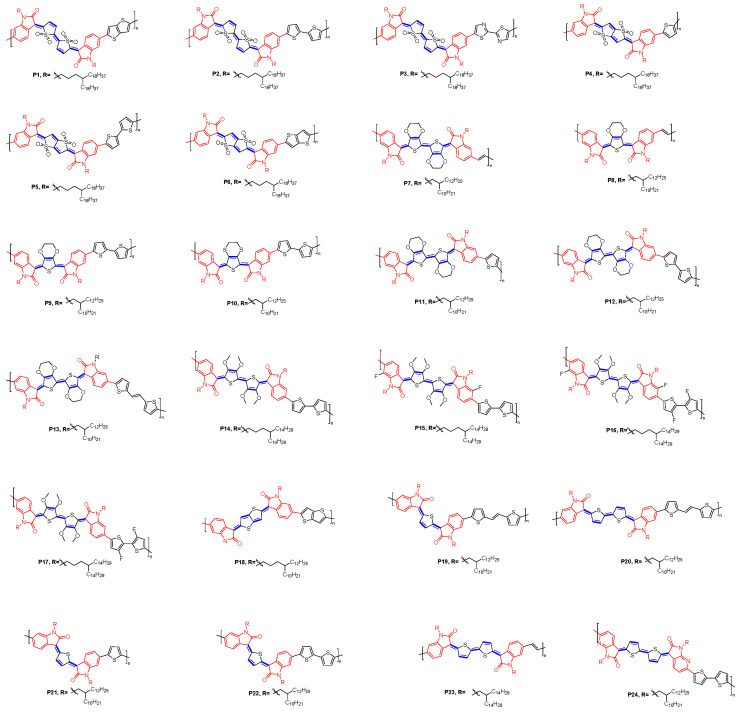
Chemical structures of copolymers based on indophenine dyes reported in the literature.

**Figure 16 materials-16-02474-f016:**
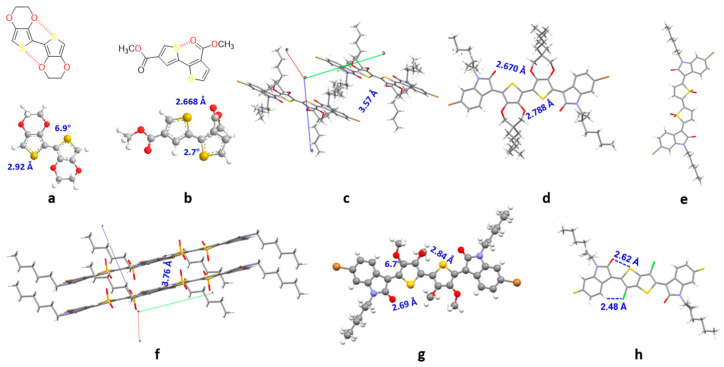
Representative examples of indophenine dyes with different conjugated π-bridge cores showing conformational locking. Adapted with permission from references (**a**,**b**) O⋯S non-covalent interactions examples [58]. Copyright (2001), *Tetrahedron Lett.* (**c**,**d**) Single crystal structure of compound **16** [52]. Copyright (2019), *J. Org. Chem.* (**e**,**f**) Single crystal structure of compound **9c** [46]. Copyright (2016), *Angew. Chem. Int. Ed.* (**g**) Single crystal structure of compound **20a** [56]. Copyright (2020), *J. Mater. Chem. C.* (**h**) Single crystal structure of compound **15b** [51]. Copyright (2022), *Chem. Eur. J.*

**Figure 17 materials-16-02474-f017:**
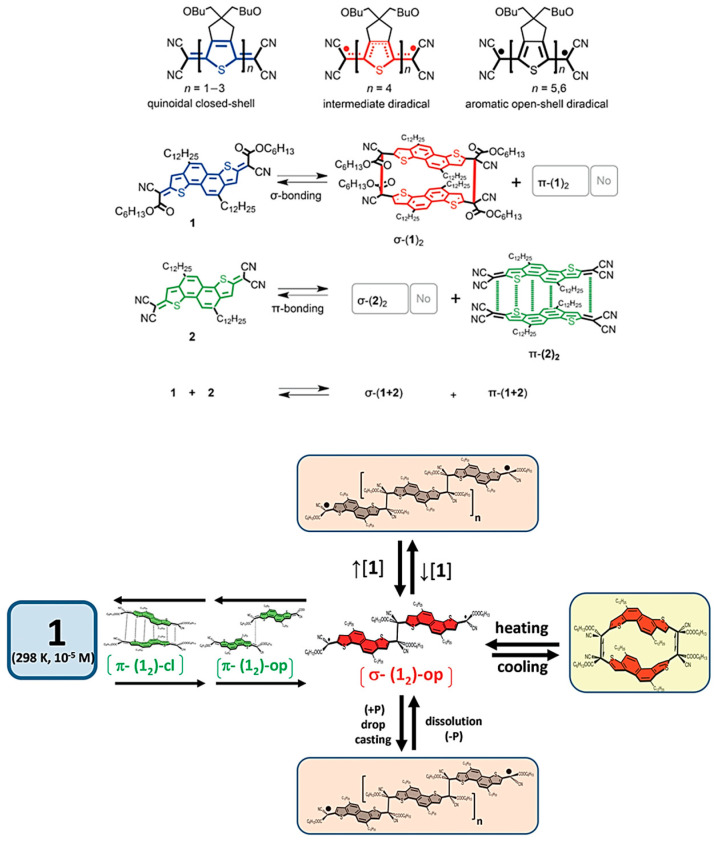
Left: Multiple di-radical properties and combinations of forms of oligothiophene. Right: multiple conformations and modes of dimerization reactions. Adapted with permission from references [67]. Copyright (2016), *Angew. Chem. Int. Ed*.

**Figure 18 materials-16-02474-f018:**
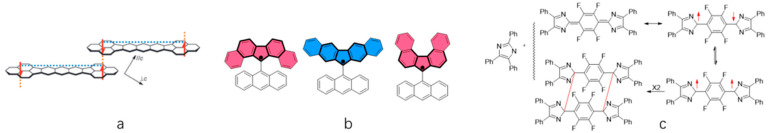
(**a**) Intramolecular and intermolecular forces present in Kekulé molecules. Adapted with permission from references. Adapted with permission from references [69]. Copyright (2012), *Chem. Commun*. (**b**) Three isomers of fluorenyl—naphthalene-1, -2 and -3, based radicals stabilized by stabilized by 9-anthryl group [70]. Copyright (2014), *J. Am. Chem. Soc.* (**c**) Photochromic Kekulé hydrocarbon with an imidazolyl radical moiety [71]. Copyright (2013), *Chem. Rev*.

**Figure 19 materials-16-02474-f019:**
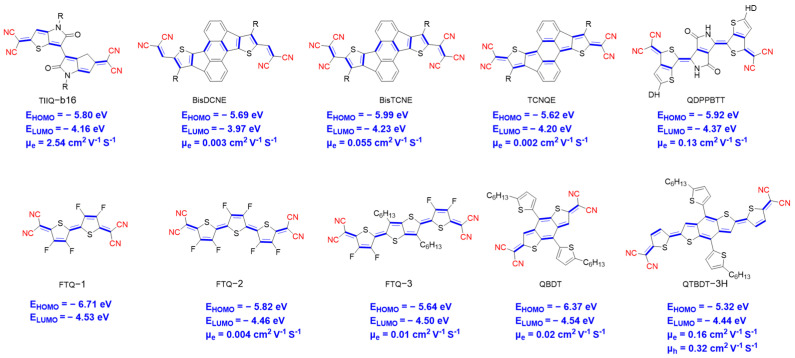
Example of molecular materials, based on dicyanomethylene end-capped OSCs for OFET application.

**Figure 20 materials-16-02474-f020:**
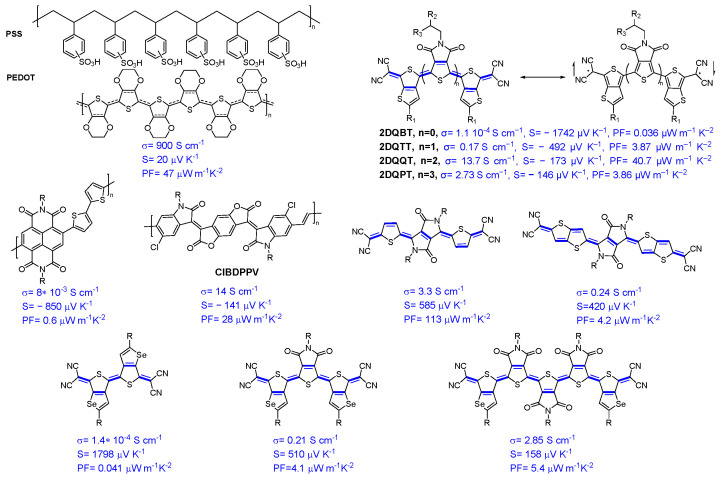
Top: Chemical structure of PEDOT:PSS, resonance structures, and diradical dicyanomethylene end-capped oligothiophene 2DQBT-2DQPT. Adapted with permission from references [87]. Copyright (2019), *Chem*. Middle: Chemical structure of CIBDPPV [86]. Copyright (2017), *Adv. Mater*. Bottom: Chemical structures of the most performing p-, n-type OSCs, alongside with diradical materials developed for TE applications [88]. Copyright (2019), *Angew. Chem. Int. Ed*.

**Figure 21 materials-16-02474-f021:**
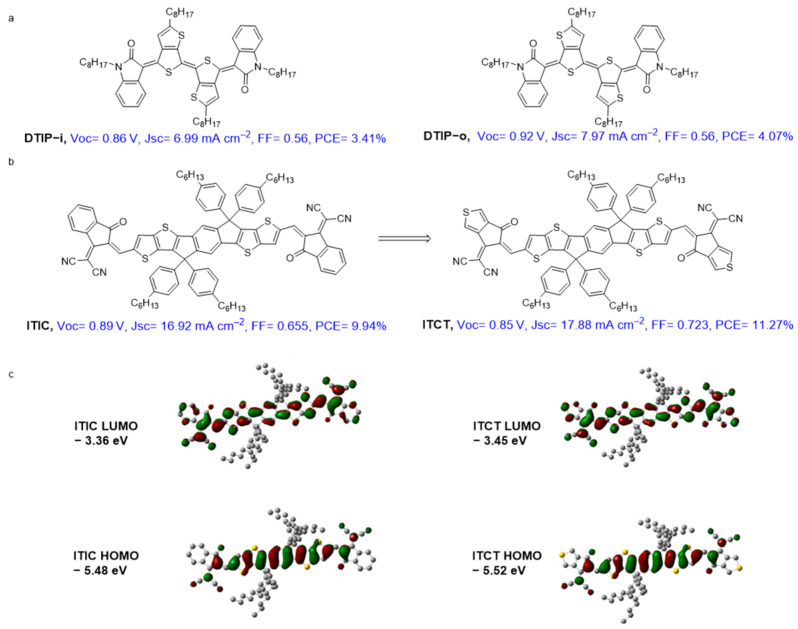
(**a**) Molecular structures of quinoidal DTIP-i and DTIP-o; (**b**) Molecular structures of ITIC and ITCT; (**c**) LUMO and HOMO energy levels distributions. Reproduced with the permission from ref [89]. Copyright (2017), *Chin. Chem. Lett.*

**Table 1 materials-16-02474-t001:** Molecular optical properties and electrochemical characteristics of small molecular based on indophenine reaction derivatives.

Materials	λ_max_^sol^ (nm)	λ_max_^film^ (nm)	E_g_ ^opt-film^ (eV)	E_HOMO_ (eV)	E_LUMO_ (eV)	E_g_ (eV)
**8**	600	650	1.55	−5.41	−3.86	1.55
**9**	450	500	1.83	−5.91	−4.08	1.83
**10c**	479	497	1.67	−6.33	−4.12	2.21
**11a**	476	491	1.74	−5.85	−3.99	1.86
**11b**	482	507	1.72	−6.30	−4.18	2.12
**13a**	427	423	1.81	−5.98	−4.17	1.81
**13b**	431	427	1.85	−6.10	−4.25	1.85
**13c**	430	425	1.82	−6.14	−4.32	1.82
**14**	665	618	1.56	−5.10	−3.58	1.52
**15a**	542	467	1.65	−5.47	−3.97	1.54
**15b**	606	475	1.63	−5.53	−4.01	1.56
**15c**	620	486	1.56	−5.58	−4.05	1.53
**16**	650	-	1.44	−5.30	−3.86	1.44
**18**	537	-	1.95	−5.29	−3.47	1.82
**19**	592	-	1.79	−5.33	−3.73	1.60
**21a**	633	-	1.67	−5.10	−5.39	1.71
**21c**	661	-	1.55	−4.85	−3.40	1.45

**Table 2 materials-16-02474-t002:** Optical properties and energy levels of conjugated copolymers based on indophenine dyes.

Materials	λ_max_^sol^ (nm)	λ_max_^film^ (nm)	E_g_ ^opt-film^ (eV)	E_HOMO_ (eV)	E_LUMO_ (eV)	E_g_ (eV)
**P1**	710	710	1.42	−5.92	−3.98	1.94
**P2**	780	-	1.43	−5.78	−4.09	1.69
**P3**	636	-	1.62	−5.99	−4.18	1.81
**P4**	740	723	1.48	−6.00	−4.04	1.96
**P5**	751	740	1.47	−5.95	−4.03	1.92
**P6**	758	750	1.43	−5.91	−4.05	1.86
**P7**	811	821	1.17	−4.59	−3.62	0.96
**P8**	753	748	1.52	−5.14	−3.62	1.52
**P9**	670	743	1.52	−5.17	−3.58	1.59
**P10**	723	787	1.36	−5.17	−3.70	1.47
**P11**	787	794	1.15	−4.58	−3.43	1.15
**P12**	746	752	1.16	−4.52	−3.36	1.16
**P13**	728	712	1.18	−4.50	−3.32	1.18
**P14**	760	758	1.18	−5.31	−3.95	1.36
**P15**	984	965	1.15	−5.35	−3.92	1.43
**P16**	936	930	1.13	−5.43	−3.97	1.46
**P17**	892	794	1.14	−5.32	−3.92	1.40
**P18**	876	910	1.28	−5.19	−3.63	1.56
**P19**	728	778	1.34	−5.22	−3.43	1.79
**P20**	774	900	1.16	−5.04	−3.65	1.39
**P21**	768	773	1.44	−5.24	−3.56	1.68
**P22**	773	769	1.45	−5.11	−3.55	1.56
**P23**	807	829	1.12	−5.08	−3.79	1.29
**P24**	845	981	1.11	−5.03	−3.74	1.29

**Table 3 materials-16-02474-t003:** OFET performance of polymers used in transistor applications (n-type).

Materials	λ_max_ (nm)	E_g_ ^opt^ (eV)	E_HOMO_ (eV)	E_LUMO_ (eV)	OFET structure	μ_e_ (cm^2^/V·s)	Measured Environment
**P1**	710	1.42	−5.92	−3.98	BGBC	0.14	N_2_
**P2**	765	1.43	−5.78	−4.09	BGBC	0.18	N_2_
**P3**	740	1.62	−5.99	−4.18	BGBC	0.016	N_2_
**P4**	740	1.48	−6.00	−4.04	TGBC	0.004	air
**P5**	751	1.47	−5.95	−4.03	TGBC	0.38	air
**P6**	758	1.43	−5.91	−4.04	TGBC	0.45	air

λ_max_: maximum absorption wavelength; E_g_^opt^: the optical energy gap estimated from the absorption onset = 1240/λ_onset_; BGBC: bottom gate bottom contact OFETs devices structures; TGBC: top gate bottom contact OFETs devices structures; μ_e_: electron mobilities.

**Table 4 materials-16-02474-t004:** OFET device performance of polymers used in transistor applications (p-type).

Polymers	E_HOMO_ (eV)	E_LUMO_ (eV)	OFET Structure	μ_h_ (cm^2^/V·s)	μ_e_ (cm^2^/V·s)	Measured Environment
**P7**	−4.59	−3.63	TGBC	0.12		N_2_
**P8**	−5.14	−3.62	TGBC	0.024	0.049	N_2_
**P9**	−5.17	−3.58	TGBC	0.11	0.0014	N_2_
**P10**	−5.17	−3.70	TGBC	0.017	0.00092	N_2_
**P11**	−4.58	−3.43	TGBC	0.043		N_2_
**P12**	−4.52	−3.36	TGBC	0.018		N_2_
**P13**	−4.50	−3.32	TGBC	0.007		N_2_
**P14**	−5.31	−3.95	TGBC	0.10		N_2_
**P15**	−5.35	−3.92	TGBC	0.91		N_2_
**P16**	−5.43	−3.97	TGBC	2.70		N_2_
**P17**	−5.32	−3.92	TGBC	1.35		N_2_
**P18**	−5.19	−3.63	BGTC	0.13		N_2_
**P19**	−5.22	−3.43	TGBC	2.40	0.056	N_2_
**P20**	−5.04	−3.65	TGBC	0.055	0.0015	N_2_
**P21**	−5.24	−3.56	TGBC	4.82	1.11	N_2_
**P22**	−5.11	−3.55	TGBC	8.09	0.74	N_2_
**P23**	−5.08	−3.79	TGBC	0.52	0.53	N_2_
**P24**	−5.03	−3.74	TGBC	0.35	0.46	N_2_

BGTC: bottom gate top contact OFET device structure; TGBC: top gate bottom contact OFET device structure; μ_h_: hole mobility; μ_e_: electron mobility.

## Data Availability

Not applicable.

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
