# Peer review of "Recent Research Progress in Indophenine-Based-Functional Materials: Design, Synthesis, and Optoelectronic Applications"

_materials, 2023, doi:10.3390/ma16062474_

Round 1

Reviewer 1 Report

This review, “Recent Research Progress in Quinoidal and Indo-phenine-Based-Functional Materials: Design, Synthesis, and Optoelectronic Applications”, discusses recent developments in the design, synthesis, properties, and device performance of quinoidal π-conjugated materials, with a focus on emerging materials like indophenine dyes that have potential for high-performance devices. The review provides an introduction to organic electronic devices, such as thin-film transistors, solar cells, and organic thermoelectric generators, and outlines key performance parameters. The authors review recent advances in the synthesis of quinoidal π-conjugated semiconducting materials, with a focus on the indophenine family. They discuss the relationship between molecular structure, energy levels, and properties in these materials, and explore the effect of structural modifications on device performance. The review aims to provide a general understanding of the design principles for synthesizing new generations of quinoidal or diradical materials with tunable optoelectronic properties and improved charge carrier mobility. The topics are well-organized and thoroughly discussed. I would recommend it be published in Materials after the following minor issues are addressed:

  1. To provide guidance for designing stable electron transport characteristics, it would be beneficial to include a brief paragraph on the stability of chemical materials and devices. This could aid in designing chemical structures that are more stable and reliable in the long-term.
  2. In order to inspire further research in this dynamic field, the authors could consider sharing their personal perspectives on emerging trends and outstanding issues. This would provide valuable insights for researchers looking to explore new avenues and address current challenges in the field.

Author Response

Reviewer #1

Point 1: To provide guidance for designing stable electron transport characteristics, it would be beneficial to include a brief paragraph on the stability of chemical materials and devices. This could aid in designing chemical structures that are more stable and reliable in the long-term.

Response: Thank you for your pertinent suggestion. We have added a paragraph in page 22, line 4385.

N-type OSC-materials are less developed than their p-type counterpart, due to their instability in air as well as their low electron mobility. So, design of n-type OSC materials with high electron mobility is an emerging field. Although, high performance OFETs device relies on not only the properties of n-type OSC active layer, but also the dielectrics and device processing techniques. An ideal n-type OSC semiconductors with high electron mobility and good stability in air should have a planar molecular structure with short intermolecular stacking distances. In addition, they should possess a deep LUMO (<-4.0 eV) to inhibit air oxidation, thus resulting in excellent ambient stability.

Point 2. In order to inspire further research in this dynamic field, the authors could consider sharing their personal perspectives on emerging trends and outstanding issues. This would provide valuable insights for researchers looking to explore new avenues and address current challenges in the field.

Again, thank you for your kind remark, we have added our own vision to the field, by summarizing the recent trends. Page 25 to page 26.

Reviewer 2 Report

The authors presented a review paper on indophenine-based functional materials which focuses on the design, synthesis and optoelectronic applications.

1) Please delete " quinodal" from the title, as this research focuses more on indophenin and not on a larger group of quinoidal structures beyond the indophenin portion.

2) The structure of the two thiophenes in Figure 3 should be realigned. The S's of the thiophene are on opposite sides and not on the same side.

3) There are a lot of typo's for instance scoop (line 16 and line 151) should be scope and so on. Please check the whole manuscript.

4) Please rephrase Line 48 "affording handle(s) for incorporation" is a misleading and confusing phrase

5) In the label of Figure 1, so that the readers will not be confused, please put hyphen or a separated to the numbers with their respective items, also please follow proper separation of text by adding punctuation marks.

6) Please rephrase the statement in line 68 just to focus on the indophenin and not the whole quinoidal pi-conjugated systems 

7) The references comma should be included in the superscript and not separated (see line 80)

8) In Figure 2, in order to guide authors it is better to put letters to indicate what they are referring to. In addition, did the authors draw the figure themselves or is it a collection from different sources. Proper citations and permission requests should be also included. This is also recommended to all figures that they present.

9) I suggest to rearrange sections 3.1 to 3.3 and expand the discussion in section 3.4 and I suggest to focus on the indophenin, since there are a lot of reviews that are even more details which regards to the other structures they mentioned. Start with simple synthesis of indophenine, then its derivatives.

10) I suggest also include challenges and future directions on these indophenine compounds.

11) There are some articles also that needs mentioning, the DOIs are as follows:

10.1016/j.polymer.2020.123032
10.1021/acsapm.2c01429
10.1021/acs.macromol.9b00370
10.1007/s11426-021-9991-0

Author Response

Review Report 2:

Point 1. Please delete " quinoidal" from the title, as this research focuses more on indophenine and not on a larger group of quinoidal structures beyond the indophenin portion.

Thanks to your suggestion, we have streamlined the scope of the article title and changed it to: Recent Research Progress in Indophenine-Based-Functional Materials: Design, Synthesis, and Optoelectronic Applications.

Point 2. The structure of the two thiophenes in Figure 3 should be realigned. The S's of the thiophene are on opposite sides and not on the same side.

We apologize for not noticing the details of the structural formula and have redrawn the structure of the chemical formula in Figure 3.

Point 3. There are a lot of typo's for instance scoop (line 16 and line 151) should be scope and so on. Please check the whole manuscript.

Again we apologize for some spelling errors, we have rechecked every spelling word by word.

Point 4. Please rephrase Line 48 "affording handle(s) for incorporation" is a misleading and confusing phrase

We have re-written line 48 to make it clearer and easier to understand.

Point 5. In the label of Figure 1, so that the readers will not be confused, please put hyphen or a separated to the numbers with their respective items, also please follow proper separation of text by adding punctuation marks.

Thank you for your pertinent advice, hopefully the title is now less likely to be misleading. It now stands as: The number of review on the topic n-type OSCs from Thomson Reuters Web of Science: 114, of which the number of review: 114 and Quinoidal OSCs reviews: 9. Of the indophenine 106 references no reviews article reported according to Scfinder.

Point 6. Please rephrase the statement in line 68 just to focus on the indophenin and not the whole quinoidal pi-conjugated systems

Thank you for your comment, it has been modified accordingly in the statement on line 68, replace quinoidal with indophenine.

Point 7. The references comma should be included in the superscript and not separated (see line 80)

Thank you for your attention to this detail, which has been corrected. I have checked all similar issues and made changes.

Point 8. In Figure 2, in order to guide authors it is better to put letters to indicate what they are referring to. In addition, did the authors draw the figure themselves or is it a collection from different sources. Proper citations and permission requests should be also included. This is also recommended to all figures that they present.

Thank you for your useful suggestion.

We labeled on the figure, a; b; c; and d, and explained in the legend to the corresponding function. We have drawn all the structures reported in figure2. Such structures are common in the literature and do not require copyright permission. Other figures form literature are produced with copyright permission form editors.

Point 9. I suggest to rearrange sections 3.1 to 3.3 and expand the discussion in section 3.4 and I suggest to focus on the indophenin, since there are a lot of reviews that are even more details which regards to the other structures they mentioned. Start with simple synthesis of indophenine, then its derivatives.

Thank you for your suggestion, and indeed as you mentioned, the article should devote to indophenine based material discussing key molecular design startaegy and structure-properties relationship. To do so, we employ some typical structures of quinoidal non-indophenine materials, that are provided in the three paragraphs 3.1-3.3, to facilitate the reader's understanding of indophenine materials and to compare them with the other three classes of materials in terms of structure, properties and synthetic methods. The design aspects of these materials, such as the investigation of intermolecular interactions and the investigation of di-radicals, are echoed in subsequent sections of Part IV. We would prefer to use the first three classes of materials as a lead-in to a more detailed discussion of class of indophenine materials.

Point 10. I suggest also include challenges and future directions on these indophenine compounds.

Thanks to your suggestions, we have rewritten the conclusion, page 25 to page 26; by addressing the challenges and perspectives.

Point 11. There are some articles also that needs mentioning, the DOIs are as follows:

10.1016/j.polymer.2020.123032

10.1021/acsapm.2c01429

10.1021/acs.macromol.9b00370

10.1007/s11426-021-9991-0

Thank you for sharing with us the messing references. We have added these four articles as references to this article.

10.1016/j.polymer.2020.123032, We added this reference and discussed the location of the quinoidal structure. (ref 55)

10.1021/acsapm.2c01429, In fact this ref was discussed in the original manuscript in reference 67. 10.1021/acs.macromol.9b00370, This ref was actually discussed in the original manuscript in reference 97 as follows: Huang et al. investigated new quinoidal acceptor building blocks by incorporating a quinoidal thieno…...

10.1007/s11426-021-9991-0. This reference has been added in section 3.2 and the specific number of the article is 39.

Reviewer 3 Report

The authors provided a review paper that highlights the design, synthesis, properties, and device performance of quinoidal π-conjugated materials. Special attention was given to emerging materials like indophenine dyes. The review contains useful information about the recent state-of-the-art progress and some guidelines for the design of quinoidal organic semiconducting materials. Here are some comments to improve the work:

1. Some figures need to be revised. Examples:

Figure 1: the fonts should be enlarged. 

Figure 2 and Figure 6 and Figure 17: the size of the devices and structures should be enlarged to be clearer.

Figure 7: should be divided into two figures to give a clear view of the figures included.

2. The conclusion should be rewritten to include more information especially the authors recommendations and suggestions.

3. The text should be proofread to minimize typographical and grammatical errors.

Author Response

Reviewer #3:

Point 1. Some figures need to be revised. Examples:

Figure 1: the fonts should be enlarged.

Figure 2 and Figure 6 and Figure 17: the size of the devices and structures should be enlarged to be clearer.

Figure 7: should be divided into two figures to give a clear view of the figures included.

Thank you for the reminder. We have restructured these five figures to make them clearer and easier for the reader to read. Figure 1 has been enlarged, the composition of Figure 2 (with added numbering) and the position of the structural formula in Figure 4 have been adjusted separately, and Figure 17 has been further scaled to make it more visible. Figure 7 has been divided into two parts and the size of each part has been increased.

Point 2. The conclusion should be rewritten to include more information especially the authors recommendations and suggestions.

Thanks to your comments, we have rewritten the conclusion section and we added more information and present our suggestions and ideas for research on the topics. Page 25 to page 26.

Point 3. The text should be proofread to minimize typographical and grammatical errors.

We apologise for the typographical and grammatical errors, we have carefully proofread the text and thank you for your corrections.

Round 2

Reviewer 2 Report

The authors have answered/replied to all of the comments/suggestions of the reviewer. But there are still few suggestions they need to address before it can be accepted for publications.

1) Figure 16 - please remove the borders of the figure.

2) When citing references, please follow the following: 

Adapted with permission from reference [1]. Copyright (2019), Nature.

Adapted with permission from reference [2]. Copyright The Authors, some rights reserved, exclusive licensee MDPI. Distributed under a Creative Commons Attribution License 4.0 (CC BY)  https://creativecommons.org/licenses/by/4.0/).

3) In table 4, please include the definition of the abbreviations

4) No reference for Figure 20

Author Response

The authors have answered/replied to all of the comments/suggestions of the reviewer. But there are still few suggestions they need to address before it can be accepted for publications.

Thank you for all your comments and suggestions on the manuscript, which are vital to the quality of this article, for your valuable time and for the effort you have made.

We have made change in the revised manuscript (second round). The changes are  highlighted in green.

Point 1) Figure 16 - please remove the borders of the figure.

We have modified this Figure and checked all Figure 1-21 to avoid similar problems.

Point 2) When citing references, please follow the following:

Adapted with permission from reference [1]. Copyright (2019), Nature.

Adapted with permission from reference [2]. Copyright The Authors, some rights reserved, exclusive licensee MDPI. Distributed under a Creative Commons Attribution License 4.0 (CC BY) https://creativecommons.org/licenses/by/4.0/).

Thank you for promptly pointing this out and we have amended the labelling in the figures accordingly. As follows: Figure 7+16+17+18+21 modified.

Point 3) In table 4, please include the definition of the abbreviations

We have explained Table 4 and, similarly, supplemented Table 3 to further improve the readability of the manuscript.

Point 4) No reference for Figure 20

Thank you for pointing out the problem. Corresponding changes have been made, corresponding references have been added, and copyright has been obtained.
